# Hunga Tonga-Hunga Ha'apai Volcano Impact Model Observation Comparison (HTHH-MOC) Project: Experiment Protocol and Model Descriptions

Yunqian Zhu[1,2], Hideharu Akiyoshi[3], Valentina Aquila[4], Elisabeth Asher[1,5], Ewa M. Bednarz[1,2], Slimane Bekki[6],  Christoph Brühl[7], Amy H. Butler[2], Parker Case[8], Simon Chabrillat[9], Gabriel Chiodo[10], Margot Clyne[11,12], Peter R. Colarco[8, 37], Sandip Dhomse[18], Lola Falletti[6], Eric Fleming[8,13], Ben Johnson[38], Andrin Jörimann[10,36], Mahesh Kovilakam[15], Gerbrand Koren[16], Ales Kuchar[17], Nicolas Lebas[14], Qing Liang[8], Cheng-Cheng Liu[12], Graham Mann[18], Michael Manyin[8,13], Marion Marchand[6], Olaf Morgenstern[19,20*], Paul Newman[8], Luke D. Oman[8], Freja F. Østerstrøm[21,22], Yifeng Peng[23], David Plummer[24], Ilaria Quaglia[25], William Randel[25], Samuel Rémy[26], Takashi Sekiya[27], Stephen Steenrod[8,28], Timofei Sukhodolov[36], Simone Tilmes[25], Kostas Tsigaridis[29,30], Rei Ueyama[31], Daniele Visioni[32], Xinyue Wang[11], Shingo Watanabe[27], Yousuke Yamashita[3], Pengfei Yu[33], Wandi Yu[34], Jun Zhang[25], Zhihong Zhuo[35]

1.  Cooperative Institute for Research in Environmental Sciences (CIRES), University of Colorado Boulder, USA
2.  NOAA Chemical Sciences Laboratory, Boulder, USA
3.  National Institute for Environmental Studies, Tsukuba, Japan
4.  American University, Department of Environmental Science, Washington, DC, USA
5.  NOAA Global Monitoring Laboratory, Boulder, USA
6.  LATMOS, UVSQ, CNRS, INU, Sorbonne Université, Paris, France
7.  Max Planck Institute for Chemistry, Mainz, Germany
8.  NASA Goddard Space Flight Center, Maryland, USA
9.  Royal Belgian Institute for Space Aeronomy (BIRA-IASB), Brussels, Belgium
10. Institute for Atmospheric and Climate Science, ETH Zurich, Zurich, Switzerland; Instituto de Geociencias, Spanish National Research Council (IGEO-CSIC-UCM), Madrid
11. Department of Atmospheric and Oceanic Sciences, University of Colorado Boulder, Boulder, USA
12. LASP, University of Colorado Boulder, Boulder, USA
13. Science Systems and Applications, Inc., Lanham, MD, USA
14. LOCEAN/IPSL, Sorbonne Université, CNRS, IRD, MNHN, Paris, France
15. NASA Langley Research Center, VA, USA
16. Copernicus Institute of Sustainable Development, Utrecht University, Utrecht, Netherlands
17. BOKU University, Vienna, Austria
18. School of Earth and Environment, University of Leeds; UK National Centre for Atmospheric Science, University of Leeds, Leeds, UK
19. National Institute of Water and Atmospheric Research (NIWA), Wellington, New Zealand
20. School of Physical and Chemical Sciences, University of Canterbury, Christchurch, New Zealand
21. School of Engineering and Applied Sciences, Harvard University, Cambridge, MA, USA
22. Department of Chemistry, University of Copenhagen, Copenhagen, Denmark
23. Lanzhou University, Lanzhou, China
24. Climate Research Division, Environment and Climate Change Canada, Montréal, Canada

25. NCAR ACOM, Boulder, USA
26. HYGEOS, Lille, France
27. Japan Agency for Marine-Earth Science and Technology (JAMSTEC), Yokohama, Japan
28. University of Maryland-Baltimore County, Baltimore, MD, USA
29. Center for Climate Systems Research, Columbia University, New York, NY, USA.
30. NASA Goddard Institute for Space Studies, New York, NY, USA.
31. NASA Ames Research Center, Moffett Field, CA
32. Department of Earth and Atmospheric Sciences, Cornell University, Ithaca, NY
33. Jinan University, Guangzhou, China
34. Lawrence Livermore National Laboratory, USA
35. Department of Earth and Atmospheric Sciences, University of Quebec in Montreal, Montreal
(Quebec), Canada
36. Physikalisch-Meteorologisches Observatorium Davos and World Radiation Center, Davos,
Switzerland
37. Earth System Science Interdisciplinary Center, University of Maryland, College Park, MD
USA
38. Met Office Hadley Centre, Exeter, UK
* now at: German Meteorological Service (DWD), Offenbach, Germany
Correspondence to: Yunqian Zhu yunqian.zhu@colorado.edu

**Abstract:**

The 2022 Hunga volcanic eruption injected a significant amount of water vapor and a moderate
amount of sulfur dioxide into the stratosphere causing observable responses in the climate
system. We have developed a model-observation comparison project to investigate the evolution
of volcanic water and aerosols, and their impacts on atmospheric dynamics, chemistry, and
climate, using several state-of-the-art chemistry climate models. The project goals are: 1.
Evaluate the current chemistry-climate models to quantify their performance in comparison to
observations; and 2. Understand atmospheric responses in the Earth system after this exceptional
event and investigate the potential impacts in the projected future. To achieve these goals, we
designed specific experiments for direct comparisons to observations, for example from balloons
and the Microwave Limb Sounder satellite instrument. Experiment 1 consists of two sets of free-
running ensemble experiments from 2022 to 2031: one with fixed sea-surface temperatures and
sea-ice, and one with coupled ocean.  These experiments will help to: understand the long-term
evolution of water vapor and aerosols; quantify HTHH effects on stratospheric and mesospheric
temperatures, dynamics, and transport; understand the impact of dynamic changes on ozone
chemistry; quantify the net radiative forcings; and evaluate any surface climate impact.
Experiment 2 is a nudged-run experiment from 2022 to 2023 using observed meteorology. To
allow participation of more climate models with varying complexities of aerosol simulation, we
include two sets of simulations in Experiment 2: Experiment 2a is designed for models with
internally-generated aerosol while Experiment 2b is designed for models using prescribed
aerosol surface area density. This experiment will help to: analyze $H_2O$ & aerosol evolution;
quantify the net radiative forcings; understand the impacts on mid-latitude and polar $O_3$
chemistry as well as allow close comparisons with observations.
**1. Introduction and motivations of this project**
The Hunga Tonga-Hunga Ha'apai (HTHH) Impacts activity was established in the World
Climate Research Programme (WCRP) Atmosphere Processes And their Role in Climate
(APARC) as a limited-term focused cross-activity with a duration of three years. It aims to assess
the impacts of the 15 January 2022 Hunga volcanic eruption and produce an assessment to
document the Hunga impact on the climate system. The Hunga eruption injected an
unprecedented amount of water ($H_2O$) and moderate sulfur dioxide ($SO_2$) into the stratosphere
(Millan et al., 2022), presenting a unique opportunity to understand the impacts on the
stratosphere of a large-magnitude explosive phreatomagmatic eruption. The wide range of
satellite observations of the stratospheric water and sulfate plumes, global transport and
dispersion of volcanic materials, and unusual chemical and temperature signals are helpful in
assessing model representations of stratospheric chemistry, aerosol, and dynamics. For example,
the Aura Microwave Limb Sounder (MLS) observed ~150 Tg of water injected by the Hunga
eruption (Millan et al., 2022), which slowly decayed due to the polar stratospheric cloud (PSC)
dehydration process and stratosphere-troposphere exchange (Fleming et al., 2024; Zhou et al.,
2024). Large aerosol optical depth is observed by Ozone Mapping and Profiler Suite (OMPS)
(Taha et al., 2022), due to fast formation of sulfate (Zhu et al., 2022) and the high optical
efficiency of Hunga aerosol particles (Li et al., 2024). Unlike the stratospheric warming patterns
observed from previous large volcanic eruptions (El Chichón in 1982 and Pinatubo in 1991),
global stratospheric temperatures decreased by 0.5 to 1.0 K in the first two years following the
Hunga eruption, largely due to radiative cooling from injected water vapor (Randel et al., 2024).
Satellite observations in June, July, August 2022 reveal reduced lower stratospheric ozone ($O_3$)
over the SH midlatitudes and subtropics, with high levels near the equator, exceeding previous
variability. These ozone anomalies coincide with a weakening of the Brewer-Dobson circulation
during this period (Wang et al., 2023). Changes in stratospheric winds also influence the
mesosphere, leading to a stronger mesospheric circulation and corresponding temperature
changes (Yu et al., 2023). These observed phenomena provide a unique opportunity to test the
ability of chemistry-climate models to simulate the evolution of volcanic aerosols combined with
such a large amount of water vapor, as well as understand how volcanic water vapor and aerosols
modify radiative balances and stratospheric ozone.
The APARC HTHH Impacts activity aims to provide a benchmark analysis of the
eruption impacts so far, and projections of eruption climate impacts over the next few years. Two
multi-model evaluation projects are designed to facilitate the success of this activity: Tonga
Model Intercomparison Project (Tonga-MIP) (Clyne et al. 2024) and the Hunga Tonga-Hunga
Ha'apai Volcano Impact Model Observation Comparison (HTHH-MOC) Project (this paper).
The HTHH-MOC provides a foundation for a coordinated multi-model evaluation of global
chemistry-climate models' performance in response to the Hunga volcanic eruption. It defines a
set of perturbation experiments, where volcanic forcings—injected water vapor and aerosol
concentrations—are consistently applied across participating model members. HTHH-MOC
aims to assess how reliably global chemistry-climate models simulate the climate responses to
this unprecedented volcanic forcing. This project enhances our confidence in attributing and
interpreting observations following the Hunga eruption. The scientific questions related to the
HTHH-MOC are: How does the Hunga volcanic plumes' transport relate to or impact
stratospheric dynamics (such as Brewer-Dobson circulation, polar vortex and the Quasi-Biennial
Oscillation) and upper atmosphere? What are the chemical impacts of the Hunga eruption in the
stratosphere and mesosphere? What and how long is the radiative effect of the Hunga eruption?
Does Hunga impact the tropospheric/surface climate?
Therefore, the HTHH-MOC project is focused on evaluating global chemistry-climate
models regarding the following three science themes: (1) plume evolution, dispersion, and large-
scale transport; (2) impacts on stratospheric chemistry and the ozone layer; and (3) radiative
effect from the eruption and surface climate impacts. Besides the HTHH-MOC project, the
assessment also includes analysis of observations and models that are not global climate models.
In the following paragraph, we describe the HTHH-MOC experiment design and participating
models.
**2. Experiment Design**
There are two experiments (**Exp1** and **Exp2** detailed below) designed to fulfill the
scientific goals. Each experiment includes a set of simulations with different volcanic injections
(i.e. with and without water and/or SO2 injections), to explore the separate impacts of volcanic
water and aerosols during the post-eruption period: a) Control case (no eruption); b) $H_2O$ (~150
Tg) & $SO_2$ (0.5 Tg); c) Only $H_2O$ (~150 Tg). d) Only $SO_2$ (0.5 Tg). Simulations with the
injection of $SO_2$ only (d) are optional and designed for aerosol-focused models. The $SO_2$ and
water injections are prescribed based on Millan et al. (2022) and Carn et al. (2023). Note that
~150 Tg of water is not the injection amount but the amount retained after the first couple of
days. This is because some models form ice particles that fall out of the stratosphere due to large
$H_2O$ supersaturation during the initial injection (Zhu et al., 2022); these models will have to
inject more $H_2O$ to counterbalance the ice formation (see **Table 6**). The only requirement is that
the model should have reasonable comparison to the MLS observations for water vapor as shown
in **Figure 1**. Aside from retaining ~150 Tg of water, the water vapor enhancement should be near
10 hPa to 50 hPa, and most of the water vapor should be located between 10˚N and 30˚S by
March 2022.
The first experiment (**Exp1**) is a free-running ensemble simulation covering the period
from 2022 to 2031. The experiment has been designed to answer questions on: 1. Understanding
the long-term evolution of Hunga water vapor and aerosols in free-running models; 2.
Quantifying Hunga effects on stratospheric temperatures, dynamics, and transport; 3.
Understanding the impact of dynamic changes on ozone chemistry; 4. Quantifying the net
radiative effects; 5. Estimating surface impacts (e.g., temperature, El Niño-Southern Oscillation,
monsoon precipitation, etc.). Simulations with free-running meteorology are required to properly
understand the impacts of the eruption on atmospheric dynamics and transport processes, and the
resulting impacts of those on chemical species (e.g., ozone) and surface climate. Since coupling
of the atmosphere with ocean and land processes is required to fully simulate many aspects of the
surface impacts, the use of coupled atmosphere, ocean, and land models is recommended.
However, since such a fully interactive set up imposes additional computing requirements, an
alternative model set up with fixed sea-surface temperatures (SSTs) and sea-ice is also allowed.
In that case, the prescribed climatological SSTs and sea-ice data are obtained by averaging SST
during the past decade (2012-2021), with the same data imposed in both the $H_2O+SO_2$ (b) and
control (a) simulations. It is important to note that both initial and boundary conditions in a
model come with uncertainties, and model processes are simplified. Therefore, model
simulations are influenced by the characteristics of the model itself and the background state of
the atmospheric system (Jones et al., 2016; Brodowsky et al., 2021). To address some of the
inherent uncertainties and reduce contribution of interannual variability to the forced response,
we use a large ensemble of simulations with slightly varied initial conditions. Note that in the
projection of stratospheric volcanic forcing, we only considered the Hunga eruption since 2022,
and no future explosive eruptions are included. For example, the 2024 Mt. Ruang eruption
contributed to elevated stratospheric aerosol optical depth, but it is not included.
Particularly, the first 5 years of qualified models output of **Exp1** are used to understand
climate impacts on the mesosphere and ionosphere from 2022-2027, such as gravity wave drag,
temperature changes, polar mesospheric clouds (PMCs), and atmospheric circulation. The
qualified models need to resolve the upper atmosphere with vertical resolutions higher or equal
to what we request in **Section 3**.
Since some aspects of the response, e.g., impacts on the radiative effect, may be too noisy
from free-running model simulations even with large ensembles, we have also designed the
second experiment which uses nudged temperature and meteorology to ensure that the
meteorology will be as close as possible to the one observed and thus isolate chemical changes
and their radiative effect. Experiment 2 (**Exp2**) is a two-year simulation that runs from 2022 to
2023 with nudged winds and/or temperature to answer questions on $H_2O$ and aerosol evolution;
quantification of the net radiative effects; and impacts on mid-latitude and polar ozone
chemistry. **Exp2** has two distinct realizations: Experiment 2a (**Exp2a**) and Experiment 2b
(**Exp2b**). The models participating in **Exp2a** all have a prognostic aerosol module, but vary in
the complexity of their representation of aerosol microphysics (i.e., bulk, modal, or sectional).
Models participating in **Exp2b** use prescribed aerosol surface area density (SAD) and radiative
properties as input to the models (Jörimann et al., 2025). The prescribed aerosol properties are
calculated using Global Space-based Stratospheric Aerosol Climatology (GloSSAC; Thomason
et al., 2018; Kovilakam et al., 2020, 2023) version 2.22 aerosol data from 1979-2023. Note that
for the period after the Hunga eruption, GloSSAC uses the Stratospheric Aerosol and Gas
Experiment (SAGEIII/ISS) version 5.3 interpolated along the time axis and the Optical
Spectrograph and InfraRed Imager System (OSIRIS) version 7.3 to fill in any missing data
poleward of 60° N/S due to the unavailability of the Cloud-Aerosol Lidar and Infrared Pathfinder
Satellite Observations (CALIPSO) data since January 2022. Therefore, when conducting
analyses north/south of 60°N/S it should be noted that the aerosols may be underestimated due to
the OSIRIS instrument retrieval biases. We ask for the models to check their initial chemical
fields against MLS to see if the models are qualified to evaluate their ozone chemistry. The
nudged runs of **Exp2** enable isolation of the chemical impact of the Hunga eruption from the
volcanically induced changes in dynamics by comparing the runs with and without $H_2O+SO_2$
injection. The net radiative effect anomaly due to water and sulfate aerosol can also be calculated
by comparing the control run (a) with the $H_2O+SO_2$ injection run (b).
**Table 1** shows the forcings and emissions data used for the HTHH-MOC experiments.
**Table 2** shows the settings specific to each experiment. For volcanic injection for **Exp1** and **2**,
we recommend the injections of $H_2O$ and $SO_2$ at 4 UTC on Jan 15, 2022. All the models are
required to retain a similar amount of water as observed by MLS (~ 150 Tg). The models are
recommended to compare with the MLS evolution for validation (**Figure 1**). The goal is to retain
the same amount of water and similar altitude to start with, so we can analyze the water's impact
on the stratosphere and climate. If injecting 25-30 km cannot retain 150 Tg, models can inject
higher than 30 km. The $SO_2$ injection is required to be 0.5 Tg for all models. The injection
locations are not required to be co-injected with $H_2O$.
The data analysis of this project is designed to do inter-model comparisons, as well as
inter-experiment comparisons. For example, the comparisons between **Exp2a** and **Exp2b** can
help to understand how well we simulate the sulfate SAD and the importance of SAD variation
for stratospheric ozone chemistry. Comparing **Exp1** and **Exp2** for the same period can help
understand instantaneous and adjusted radiative effects. In addition, large (10-20) member
ensembles are requested for free-running simulations to better quantify the role of internal
variability in the climate response.
**Table 1. Summary of forcings and emissions data used in HTHH-MOC experiments.**

| Spin-up* | 5 years nudged runs |
|---|---|
| Degassing** and eruptive volcano source | Need both degassing and eruptive volcanic input for 5-year spin-up. Degassing continues during the experiment runs (e.g. 10 years for **Exp1**, 2 years for **Exp2**). recommended references: Volcanic degassing Carn et al. (2017); Eruptive volcanoes (Neely III, & Schmidt (2016) https://archive.researchdata.leeds.ac.uk/96/ or Carn et al. (2017); Assume no more explosive volcanoes after Hunga. |
| Surface emission | Coupled Model Intercomparison Project phase 6 (CMIP6) emissions follow SSP2-4.5 (Gidden et al., 2019), which adopts an intermediate greenhouse gas (GHG) emission: $CO_2$ emissions around current levels before beginning to decline by 2050. |
| Chemical initialization | Stratospheric chemistry fields (such as $O_3$, $H_2O$) at the beginning of 2022 should be compared with MLS observations for validation if the model participates in evaluation of the Hunga stratospheric chemistry impact. |

* 5 years is enough to reach sulfate equilibrium in the stratosphere; water may take 7 years (each model
should adjust the spin-up time according to model features). ** Recommended degassing volcanic
emissions injected at the cone altitude, constant flux based on Carn et al. (2017). Database is updated
through 2022 here: https://doi.org/10.5067/MEASURES/SO2/DATA406.
**Table 2. Experiment design**

| Experiment | Meteorology | period | aerosol treatment | QBO | SST | Ensemble members |
|---|---|---|---|---|---|---|
| Exp1.FixedSST | | | | Internal generated (Nudge if model doesn't generate) | Fixed (climatology = mean of monthly average during the past decade (2012-2021), repeating annually) This applies to spin-up time too. | 10-20 |
| | Free run starts Feb 1. (i.e. nudge until Jan 31) | 10 years 2022-2031 (first 5 years for mesospheric analysis) | model simulated aerosol or prescribed | | | |
| Exp1.Coupled Ocean | | | | | Coupled ocean (optional) initialize with observed ocean state (see section 3 for individual model descriptions) | 10-20 |
| Exp2a | Nudged wind only and/or nudged T and wind* | 2 years 2022-2023 | model simulated aerosol | nudged | Observed SST | 1 |
| Exp2b | Nudged wind only and/or | 2 years 2022-2023 | prescribed | nudged | Observed SST | 1 |


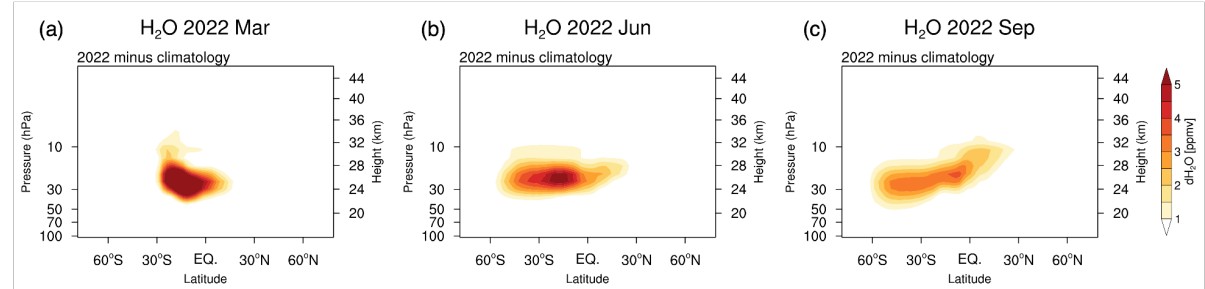

**Figure 1.** Monthly average water vapor perturbation after the Hunga eruption from MLS. Panels
(a-c) show the observed dispersion of the $H_2O$ enhancement in 2022 in the months of (a) March,
(b) June, and (c) September.

A parallel model intercomparison project Tonga-MIP (Clyne et al., 2024) will also be
part of the 2025 Hunga assessment, which is designed to explore the plume evolution between 1
day and up to 1 or 2 months after the eruption. Tonga-MIP was initiated before the APARC
Hunga Activity started. It will be described in a separate paper, but we list it in this paper to
document the comprehensiveness of the modeling effort for the Hunga assessment. Two
purposes of Tonga-MIP cannot be achieved by Exp1 and 2: 1. The nudged experiment of Tonga-
MIP aims to intercompare the microphysics processes (i.e., cloud and aerosol physics and sulfur
chemistry) between different models. Therefore, all models are requested to inject 150 Tg of
water, but the retaining of the water varies between models, differing from Exp1 and 2, which
ask to retain ~150 Tg of water in the stratosphere. SO2 injection is 0.5 Tg, the same as
experiments in HTHHMOC. The injections are required to be injected between 25-30 km, within
the latitude and longitude box of 22-14°S and 182-186°E, at a constant vertical volume mixing
ratio for 6 hours starting at 4 UTC on January 15th. 2. The free-run experiment of Tonga-MIP
aims to study the radiative effect of water and $SO_2$ on the Hunga plume descending and
ascending during the first month after the eruption since the Hunga water and aerosol plumes
were observed to descend several kilometers during the first monthly after the eruption (Sellito et
al., 2022; Randel et al., 2024). Therefore, Tonga-MIP designed to nudge the atmosphere up until
several different dates and explore the plume descending patterns with free-run atmosphere after
these dates. The dates are Jan 21, Jan 26 and Jan 31. Most of the models that participate in
Tonga-MIP also participate in the HTHH-MOC.


**3. Model output**
The model output covers variables based on the Chemistry-Climate Modeling Initiative
(CCMI) output list with some additions specific to this study. The detailed list is provided in the
**Supplementary Excel Table**. We have requested that all models generate the same variable
names, units, ordering of dimensions (longitude from 0˚E to 360˚E; latitude from 90˚S to 90˚N;
pressure levels from 1000 hPa to 0.03 hPa or altitude from 0 meter to 85,000 meter), and file
name structure (e.g. 'variable_domain_modelname_experimentname.nc' or
'domain_modelname_experimentname.variable.nc'). The examples of Experiment_name are:
HTHHMOC-Exp1, HTHHMOC-Exp1. The example file names are:
Monthlymean_WACCM6MAM_HTHHMOC-Exp1-NoVolc-fixedSST.ensemble001.O3.nc or
O3_Dailymean_WACCM6MAM_HTHHMOC-Exp1-H2Oonly-CoupledOcean.ensemble001.nc.
The 3D model output is requested on both model levels (hybrid pressure or height) and
interpolated to CMIP6 plev39 grid (plev39: 1000, 925, 850, 700, 600, 500, 400, 300, 250, 200,
287 170, 150, 130, 115, 100, 90, 80, 70, 50, 30, 20, 15, 10, 7, 5, 3, 2, 1.5, 1 0, 0.7, 0.5, 0.4, 0.3, 0.2,
0.15, 0.1, 0.07, 0.05, 0.03 hPa) and for mesospheric analysis adding 0.02, 0.01, 0.007, 0.005,
0.003, 0.001 above the plev39 grid.
Monthly mean output is requested for all variables for **Exp1** with some fields (specified
in the Excel sheet) as daily mean. Some of the fields requested as daily means are specified,
either as surface fields or at reduced number of pressure levels. Daily mean output is requested
for all variables for **Exp2**.
The model output (~33 TB) **of Exp1 and Exp2** is archived at the JASMIN workspace
(jasmin.ac.uk). JASMIN provides large storage space and compute facilities to facilitate the data
archiving and post data analysis of this project. This reduces the need for data transfers and
allows reproducible computational workflows. Seddon et al. (2023) described the facility in
detail. Our next phase is to publicly release the data by transferring the data to the Centre for
Environmental Data Analysis (CEDA) archiving system.
**4. Model Descriptions and the Hunga Volcanic Injection Specification**
As part of the three-year Hunga Impact activity, this project is highly time-sensitive. We
designed the timeline for each experiment (**Figure 2**) to facilitate the completion of the 2025
Hunga Impact assessment. However, the JASMIN workspace will remain open for the uploading
of modeling data after the deadline denoted in **Figure 2** until 2025**.**
This paper only includes model descriptions for those models that submitted the output
following the assessment timeline. The model setup follows the protocols listed in Section 2
unless specified below. **Tables 3-6** provide key information on the participant models, which are
detailed described in the following paragraphs for each model.

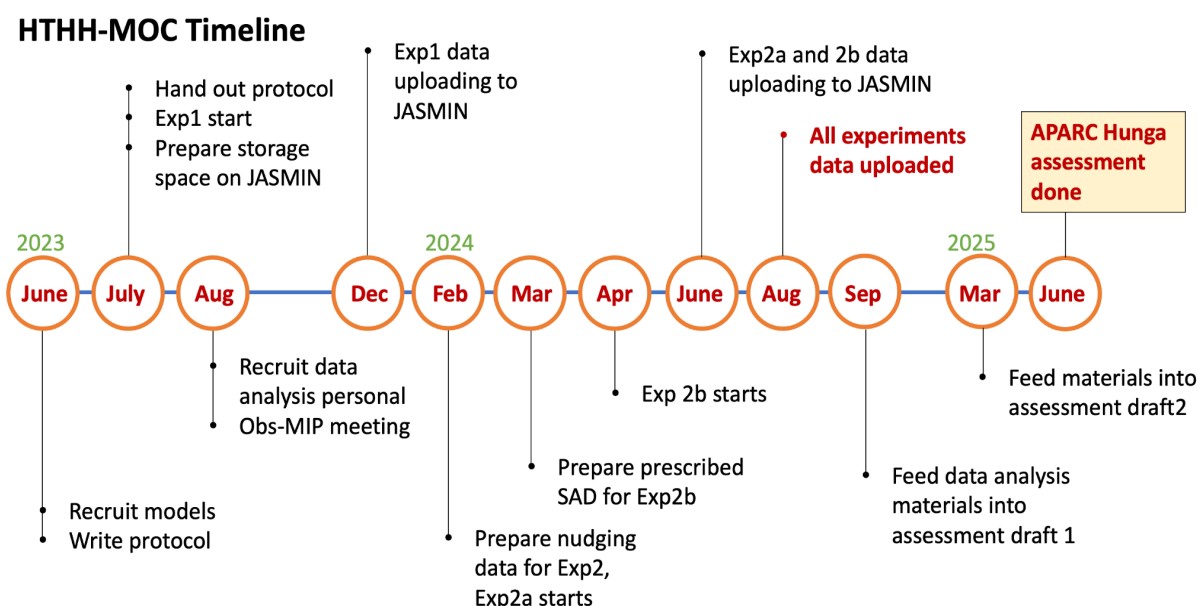


**Figure 2.** The timeline designed for HTHH-MOC in order to cooperate with the APARC HTHH
Impact assessment.

**Table 3. Participating models and contact information for HTHH-MOC and Tonga-MIP.**

| Model name | Description reference paper | Institutions (**that develop the model**) | Primary contact (**who runs the model**) | Emails |
|---|---|---|---|---|
| CAM5/CARMA | Yu et al. (2015) | CU Boulder Jinan Univ. | Pengfei Yu Yifeng Peng | pengfei.yu@colorado.edu pengyf16@lzu.edu.cn |
| CCSRNIES-MIROC3.2 | Akiyoshi et al. (2023), Akiyoshi et al. (2016) | NIES | Yousuke Yamashita Hideharu Akiyoshi | yamashita.yosuke@nies.go.jp hakiyosi@nies.go.jp |
| CMAM | Jonsson et al. (2004), Scinocca et al. (2008) | CCCma, Environment and Climate Change Canada | David Plummer | david.plummer@ec.gc.ca |
| EMAC MPIC | Schallock et al. (2023) | MPI-C, -M, DLR | Christoph Brühl | christoph.bruehl@mpic.de |
| GA4 UM-UKCA | Dhomse et al. (2020) | Univ. Leeds | Graham Mann, Sandip Dhomse | G.W.Mann@leeds.ac.uk, S.S.Dhomse@leeds.ac.uk |
| GEOSCCM | Nielsen et al. (2017) | NASA | Peter Colarco | peter.r.colarco@nasa.gov |
| GEOS/CARMA | Nielsen et al. (2017) | NASA | Parker Case | parker.a.case@nasa.gov |
| GSFC2D | Fleming et al. (2024) | NASA | Eric Fleming | eric.l.fleming@nasa.gov |
| IFS-COMPO Cy49R1 | Huijnen et al. (GMD, 2016), Rémy et al. (GMD, 2022) | ECMWF and team CAMS2_35 | Simon Chabrillat Samuel Rémy | Simon.chabrillat@aeronomie.be sr@hygeos.com |
| LMDZ6.2-LR-STRATAER/LMDZ6.2-LR-STRATAER-REPROBUS | O. Boucher et al. 2020, Marchand et al., 2012 | CNRS, Sorbonne Univerité, IPSL, LATMOS, LOCEAN | Marion Marchand, Slimane Bekki, Nicolas Lebas, Lola Falletti | marion.marchand@latmos.ipsl.fr, slimane.bekki@latmos.ipsl.fr, nicolas.lebas@locean.ipsl.fr, lola.falletti@latmos.ipsl.fr |
| MIROC-CHASER | Sekiya et al. (2016) | JAMSTEC | Shingo Watanabe, Takashi Sekiya | wnabe@jamstec.go.jp, tsekiya@jamstec.go.jp |
| MIROC-ES2H | Tatebe et al. (2019), Kawamiya et al. (2020) | JAMSTEC and NIES | Shingo Watanabe, Takashi Sekiya, Tatsuya Nagashima, Kengo Sudo | wnabe@jamstec.go.jp, tsekiya@jamstec.go.jp, nagashima.tatsuya@nies.go.jp, kengo@nagoya-u.jp |
| SOCOLv4 | Sukhodolov et al. (2021) | PMOD/WRC and ETH-Zurich | Timofei Sukhodolov | timofei.sukhodolov@pmodwrc.ch |
| UKESM1.1 | Sellar et al. (2019, 2020), with chemistry updates from Dennison et al. (2019) | UK Met Office, UK Universities and National Centre for Atmospheric Science (NCAS) | Graham Mann, Sandip Dhomse Ben Johnson Mohit Dalvi Luke Abraham James Keeble | g.w.mann@leeds.ac.uk, s.s.dhomse@leeds.ac.uk ben.johnson@metoffice.gov.uk mohit.dalvi@metoffice.gov.uk nla27@cam.ac.uk j.keeble2@lancaster.ac.uk |

| WACCM6/CARMA | Tilmes et al. (2023) | NCAR | Simone Tilmes Cheng-Cheng Liu Yunqian Zhu Margot Clyne (Tonga-MIP) | tilmes@ucar.edu chengcheng.liu@lasp.colorado.edu yunqian.zhu@noaa.gov margot.clyne@colorado.edu |
| WACCM6/MAM | Mills et al. (2016) | NCAR | Xinyue Wang Simone Tilmes Jun Zhang Wandi Yu Zhihong Zhuo Ewa Bednarz Margot Clyne (Tonga-MIP) | xinyuew@colorado.edu tilmes@ucar.edu jzhan166@ucar.edu yu44@llnl.gov zhuo.zhihong@uqam.ca ewa.bednarz@noaa.gov margot.clyne@colorado.edu |


**Table 4. Participating models in HTHH-MOC and Tonga-MIP.**

| Model names | Exp1.FixedSST | Exp1.Coupled Ocean | Exp2a | Exp2b | Tonga-MIP (Clyne et al. 2024) |
|---|---|---|---|---|---|
| CAM5/CARMA | | | X | | |
| CCSRNIES-MIROC3.2 | | | | X | |
| CMAM | X (H2O-only) (*) | | | | |
| EMAC MPIC | | | X | | |
| GA4 UM-UKCA | | | | | X |
| GEOSCCM | X | | X | | X |
| GEOS/CARMA | | | X | | |
| GSFC2D | X (*) | | | X | |
| IFS-COMPO | | | X | | |
| LMDZ6.2-LR-STRATAER | | | X | | X |
| LMDZ6.2-LR-STRATAER-REPROBUS | | | X | | X |
| MIROC-CHASER | X | | X | | |
| MIROC-ES2H | | | | | X |
| SOCOLv4 | | | | | X |
| UKESM1.1 | | | X | | X |
| WACCM6/CARMA | | | X | | X |
| WACCM6/MAM | X(*) | X(*) | X | | X |

* The models that are qualified to analyze the mesospheric components are marked with *
symbol.

**Table 5. Model resolutions and schemes used for HTHH-MOC experiments**

| Model names | Horizontal resolution | nlevels | Model Top | Vertical resolution in the stratosphere | Aerosol scheme | Specified dynamic source | QBO for models participating free run | Chemistry package (tropospheric chemistry included?) |
|---|---|---|---|---|---|---|---|---|
| CAM5/CARMA | ~2 deg | 56 | 45 km | 1-4 km | CARMA sectional (20 bins) | GEOS5 | - | MOZART (yes) |
| CCSRNIES-MIROC3.2 | T42 | 34 | 0.01 hPa | 1-3 km | None | MERRA-2 | - | full strat; no tropo |
| CMAM | T47 | 80 | 0.0006 hPa | 0.8 - 2.5 km | None | ERA5 | nudged | stratospheric + methane-NOx in troposphere |
| EMAC MPIC | T63 | 90 | 0.01 hPa | 0.5km in LS | GMXE, modal | ERA-5 | - | MECCA, simplified troposphere |
| GEOSCCM | c90 (~1 deg) | 72 | 0.01 hPa | ~1 km | GOCART (Bulk) | MERRA-2/GEOS-FP | Internal generated | GMI (yes) |
| GEOS/CARMA | c90 (~1 deg) | 72 | 0.01 hPa | ~1 km | CARMA (sectional 24 bins) | MERRA-2/GEOS-FP | - | GMI (yes) |
| GSFC2D | 4° | 76 | .002 hPa (~ 92 km) | 1km | Prescribed only | MERRA-2 | Internal generated | full strat; partial trop |
| IFS-COMPO | $T_L$511 (~40km) | 137 | 0.01 hPa | 0.5-1.5 km | Bulk | ERA5 | - | BASCOE (strato) + CB05 (tropo) |
| LMDZ6.2-LR-STRATAER | 2.5° × 1.3° | 79 | 80km | 1-5 km | S3A(sectional 36 bins) | ERA5 | - | No |
| LMDZ6.2-LR-STRATAER-REPROBUS | 2.5° × 1.3° | 79 | 80km | 1-5 km | S3A(sectional 36 bins) | ERA5 | - | REPROBUS |
| MIROC-CHASER | T85 | 81 | 0.004 hPa | 0.7-1.2 km | MAM3 | MERRA-2 | Internal generated | troposphere-stratosphere chemistry |
| UKESM1.1 | N96 | 85 | 80km | 0.6-0.7km in LS | GLOMAP-mode | ERA-5 | Internal generated | CheST strat-trop chemistry |
| WACCM6/CARMA | ~1 deg | 70 | 140 km | 1-2 km | Sectional (20 bins) | MERRA-2 | - | MOZART (yes) |
| WACCM6/MAM | ~1 deg | 70 | 140 km | 1-2 km | MAM4 | MERRA-2 | Internal generated | MOZART (yes) |



**Table 6. Hunga volcanic injection profile for HTHH-MOC experiments**

| Model names | Data and duration | $H_2O$ amount | $H_2O$ altitude | $H_2O$ location/area | $SO_2$ amount | $SO_2$ altitude | $SO_2$ location/area |
|---|---|---|---|---|---|---|---|

| | | (left after a week) | | | | | |
|---|---|---|---|---|---|---|---|
| CAM5/CARMA | Jan 15, 6 hrs | 150 Tg (~135 Tg) | 25-35 km | 22-14°S, 182-186°E | 0.5 Tg | 20-28 km | 22-14°S, 182-186°E |
| CCSRNIES-MIROC3.2 | Jan 15, instantly | 150 Tg (~150 Tg) | 12.0-27.6 hPa | 181.4–187.0°E, 14.0–22.3°S | - | - | - |
| CMAM | Feb 20, 5 days | 150 Tg (~150 Tg) | near 25.5 km | zonally average | - | - | - |
| EMAC MPIC | Jan 16, 12hrs | 136 Tg (~130 Tg) | Gaussian centered at 21.5hPa | 23-19°S, 177-173°W | 0.4 Tg based on obs. | 23-27 km based on obs. | 30°S-5°N, 90-120°W (330°) |
| GEOSCCM | Jan 15, 6 hrs | 750 Tg (~150 Tg) | 25-30 km | 22-14°S, 182-186°E | 0.5 Tg | 25-30 km | 22-14°S, 182-186°E |
| GEOS/CARMA | Jan 15, 6 hrs | 750 Tg (~150 Tg) | 25-30 km | 22-14°S, 182-186°E | 0.5 Tg | 25-30 km | 22-14°S, 182-186°E |
| GSFC2D | use MLS $H_2O$ profile until March 1 | ~150 Tg (~150 Tg) | - | zonally average | - | - | - |
| IFS-COMPO | Jan 15, 3 hrs | 190 Tg (~150 Tg) | 25-30 km | 400 km by 200 km centered 20˚S and 175˚W | 0.5 Tg | 25-30 km | 400 km by 200 km centered 20˚S and 175˚W |
| LMDZ6.2-LR-STRATAER | Jan 15, 1 day | 150 Tg (~150 Tg) | Gaussian centered at 27.5 km and standard deviation of 2.5 km | 22°-14°S, 182-186°E | 0.5 Tg | Gaussian centered at 27.5 km and standard deviation of 2.5 km | 22-14°S, 182-186°E |
| LMDZ6.2-LR-STRATAER-REPROBUS | Jan 15, 1 day | 150 Tg (~150 Tg) | Gaussian centered at 27.5 km and standard deviation of 2.5 km | 22-14°S, 182-186°E | 0.5 Tg | Gaussian centered at 27.5 km and standard deviation of 2.5 km | 22-14°S, 182-186°E |
| MIROC-CHASER | Jan 15 4 UTC, 6 hours | 186 Tg (~150 Tg) | 25-30 km | 22-14°S, 182-186°E | 0.5 Tg | 25-30 km | 22-14°S, 182-186°E |
| UKESM1.1 | Jan 15, 6 hours | 150 Tg | 25-30km | 22-14ºS 182-186°E | 0.5Tg | 25-30km | 22-14°S, 182-186°E |
| WACCM6/CARMA | Jan 15, 6 hours | 150 Tg (~150 Tg) | 25-35km | 22-6°S,182.5-202.5°E | 0.5 Tg | 26.5-36 km | 22-6°S,182.5-202.5°E |
| WACCM6/MAM | Jan 15, 6 hours | 150 Tg | 25-35 km | 22-14°S, 182-186°E | 0.5 Tg | 20-28 km | 22-14°S, 182-186°E |

| | |
|---|---|
| | (~135 Tg) |

**4.1 CAM5/CARMA**

The atmospheric component of the Community Atmosphere Model version 5 (CAM5) (Lamarque et al., 2012) is the atmospheric component of the Community Earth System Model, version 1 (CESM1.2.2, Hurrell et al., 2013), with a top at around 45 km. CAM5 has a horizontal resolution of 1.9° latitude × 2.5° longitude, utilizing the finite volume dynamical core (Lin & Rood, 1996). The model has 56 vertical levels, with a vertical resolution ~1 km in the upper troposphere and lower stratosphere. The modeled winds and temperatures were nudged to the 3-hour Goddard Earth Observing System 5 (GEOS-5) reanalysis data set (Molod et al., 2015) every time step (30 min) by 1% (i.e., a 50 h Newtonian relaxation time scale). The aerosol is interactively simulated using a sectional aerosol microphysics model, the Community Aerosol and Radiation Model for Atmospheres (CARMA, Yu et al., 2015). The model uses the Model for Ozone and Related Chemical Tracers (MOZART) chemistry that is used for both tropospheric (Emmons et al., 2010) and stratospheric chemistry (English et al., 2011; Mills et al., 2016). The volcanic emissions from continuously degassing volcanoes uses the emission inventory RCP8.5 and FINNv1.5. No volcanic eruptions except the Hunga 2022 eruption are included.

The initial volcanic injection altitude and area are determined by validating the water and aerosol transportation in months shown in **Figure 1** following the tests in Zhu et al. (2022), Wang et al. (2023) and Zhang et al. (2024). In these simulations, the $H_2O$ is injected at 25 to 35 km altitude and $SO_2$ injected at 20 to 28 km altitude. The injection latitude ranges from 22°S to 14°S, and longitude ranges from 182°E to 186°E (Zhu et al., 2022). The initial injection of $H_2O$ is 150 Tg, with ~ 135 Tg left after the first week following the eruption.

**4.2 CCSRNIES-MIROC3.2**

The Center for Climate System Research/National Institute for Environmental Studies - Model for Interdisciplinary Research on Climate version 3.2 Chemistry Climate Model (CCSRNIES-MIROC3.2 CCM) (Akiyoshi et al. 2023) was developed based on versions 3.2 of the MIROC atmospheric general circulation model (AGCM), incorporating a stratospheric chemistry module that was developed at National Institute for Environmental Studies (NIES) and the University of Tokyo. The model has a horizontal resolution of T42 (2.8° latitude × 2.8° longitude) and 34 vertical levels, with a vertical resolution ~1 km in the lower stratosphere/upper troposphere and ~3 km in the upper stratosphere and mesosphere. The top level is located at 0.01 hPa (approximately 80 km).

The chemistry in the CCSRNIES-MIROC3.2 CCM is a stratospheric chemistry module including 42 photolysis reactions, 142 gas-phase chemical reactions and 13 heterogeneous reactions for multiple aerosol types (Akiyoshi et al., 2023). Tropospheric chemistry is not included, but the stratospheric chemistry scheme is used for both the troposphere and mesosphere.

In the CCSRNIES-MIROC3.2 CCM, only **Exp2b** can be performed. The atmospheric temperature and horizontal winds are nudged toward Modern-Era Retrospective analysis for Research and Applications Version 2 (MERRA-2) reanalysis (Gelaro et al., 2017) with a 1-day relaxation using instant values at 6-hour interval (Akiyoshi et al., 2016). The HadISST data is used during the simulation.

The CCSRNIES-MIROC3.2 CCM does not have any microphysics scheme for volcanic aerosols. The surface area and spectral optical parameters of extinction, single scattering albedo,

and asymmetric factor for Hunga aerosols were prescribed in the model from the GloSSAC version
2.22 aerosol data (Jörimann et al., 2025). $H_2O$ was injected instantly on 15 January 2022 at the 12
grids of the model in the region 181.4°E–187.0°E in longitude, 14.0°S–22.3°S in latitude, and 12.0
hPa–27.6 hPa in pressure level. A uniform number density of $1.709 \times 10^{15}$ molecules/cm$^3$ $H_2O$ was
injected in each of the 12 grids which amounts to ~150 Tg.
**4.3 CMAM**
The Canadian Middle Atmosphere Model (CMAM) is based on a vertically extended
version of CanAM3.1, the third generation Canadian Atmospheric Model (Scinocca et al., 2008).
Compared to the standard configuration of CanAM3.1, for CMAM the model top was raised to
0.0006 hPa (approximately 95 km) and the parameterization of non-orographic gravity wave
drag (Scinocca, 2003) and additional radiative processes important in the middle atmosphere
(Fomichev et al., 2004) have been included. The gas-phase chemistry includes a comprehensive
description of the inorganic Ox, NOx, HOx, ClOx and BrOx families, along with $CH_4$, $N_2O$, six
chlorine containing halocarbons, $CH_3Br$ and, to account for an additional 5 ppt of bromine from
short-lived source gases, $CH_2Br_2$ and $CHBr_3$ (Jonsson et al., 2004). A prognostic description of,
and associated heterogeneous chemical reactions on water ice PSCs (PSC Type II) and liquid
ternary solution (PSC Type Ib) particles is included, although gravitational settling
(dehydration/denitrification) is not calculated and species return to the gas phase when
conditions no longer support the existence of PSC particles.
The simulations for the HTHH-MOC simulations were performed at T47 spectral
resolution (approximately 3.8° resolution on the linear transform grid used for the model
physics), with 80 vertical levels giving a vertical resolution of approximately 0.8 km at 100 hPa,
increasing to 2.3 km above 0.1 hPa. The CMAM does not internally generate a QBO, so the
zonal winds in the equatorial region were nudged towards a dataset based on observed variations
up to December 2023, constructed using the method of Naujokat (1986) and extended into the
future by repeating a historical period that is congruent with the observed QBO in late 2023.
Water vapor from the Hunga eruption was added as a zonally average perturbation to the model
water over five days from 00 UTC on February 20, 2022. The spatial distribution of the anomaly
was designed to reproduce the water vapor anomaly observed in mid-February by the The
Atmospheric Chemistry Experiment - Fourier Transform Spectrometer (ACE-FTS) (Bernath et
al., 2005) satellite (Patrick Sheese, personal communication), with a maximum value of 13.3
ppm at 17°S and 25.5 km and producing an anomaly of ~150 Tg $H_2O$ in the stratosphere.
**4.4 EMAC MPIC**
The chemistry-climate model EMAC (ECHAM5/MESSy Atmospheric Chemistry)
consists of the European Centre Hamburg general circulation model (ECHAM5) and the
Modular Earth Submodel System (MESSy) (e.g., Jöckel et al., 2010). Here we use the version of
Schallock et al. (2023) in horizontal resolution T63 (1.87°x 1.87°) with 90 levels between the
surface and 0.01 hPa.
Vorticity, divergence, and temperatures between surface and 100 hPa are nudged to the
ERA5 reanalysis of ECMWF (Hersbach et al., 2020), as well as surface pressure. SSTs and sea
ice cover are prescribed by ERA5 data. The model can generate an internal QBO but for
comparison with observations it was slightly nudged to the Singapore data compiled by Free
University of Berlin and Karlsruhe Institute of Technology (Giorgetta et al., 2006).

The model contains gas-phase and heterogeneous chemistry on PSCs and interactive
aerosols. Surface mixing ratios of chlorine- and bromine-containing halocarbons and other long-
lived gases are nudged to Advanced Global Atmospheric Gases Experiment (AGAGE)
observations. The microphysical modal aerosol module contains four soluble and three insoluble
modes for sulfate, nitrate, dust, organic and black carbon, and aerosol water (Pringle et al.,
2010). The instantaneous radiative effect by tropospheric and stratospheric aerosols can be
calculated online by multiple calls of the radiation module. Volcanoes injecting material into the
stratosphere are considered as in Schallock et al. (2023) using the perturbations of stratospheric
$SO_2$ observed by the Michelson Interferometer for Passive Atmospheric Sounding (MIPAS) and
aerosol extinction observed by OSIRIS. This method, based typically on data of a 10-day period,
distributes the injected $SO_2$ over a larger volume than typical point source approaches using the
same integrated mass (see also Kohl et al., 2024). For Hunga this method has the disadvantage
that $H_2O$ and $SO_2$ are not co-injected since $H_2O$ is injected in 12 hours in a slab consisting of
four horizontal boxes and a Gaussian vertical distribution centered at 21.5 hPa. For **Exp2a** we
continue the 30-year transient simulation presented in Schallock et al. (2023) with and without
Hunga Tonga. The simulated $H_2O$-perturbation is consistent with **Figure 1**. The $SO_2$ injection is
derived based on the extinction from the OSIRIS observation averaged over about 10 days
(**Figure 3**) (Bruehl et al., 2023).

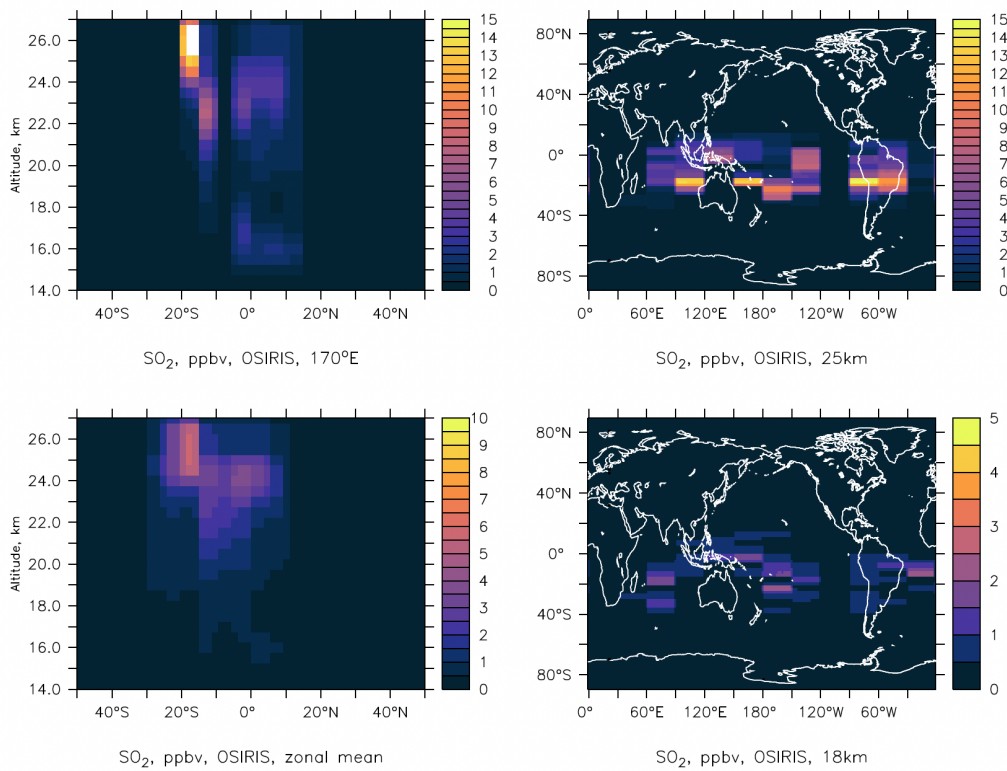

**Figure 3.** The $SO_2$ injection used in EMAC MPIC model is based on the Hunga $SO_2$
perturbation derived from extinction observed by OSIRIS averaged over about 10 days, i.e.,
including several snapshots of the westward moving plume. For conversion from extinction to
volume mixing ratio Eqn. 1 of Schallock et al (2023) is applied with f=3 because of data gaps.
5day-averaged gridded OSIRIS data averaged from January 24 0h to February 3 0h were used.
Note that the colorbars are not the same in each panel.

## 4.5 GEOSCCM

The NASA Goddard Earth Observing System Chemistry-Climate Model (GEOSCCM) is based on the GEOS Earth system model (Reinecker et al. 2008, Molod et al. 2015). For the HTHH-MOC experiments the model is run on a cubed-sphere horizontal grid at a C90 resolution (~100 km) with 72 vertical hybrid-sigma levels from the surface to 0.01 hPa (~80 km). Dynamics are solved using the finite-volume dynamical core (Putman and Lin, 2007). Deep and shallow convection are parameterized using the Grell-Freitas (2014) and Park-Bretherton (2009) schemes, respectively, and moist physics is from Bacmeister et al. (2006). The turbulence parameterization is based on the non-local scheme of Lock et al. (2000). Shortwave and longwave radiative fluxes are computed in 30 bands using the Rapid Radiative Transfer Model for GCMs (RRTMG, Iacono et al. 2008).

Stratospheric and tropospheric chemistry are from the Global Modeling Initiative (GMI) mechanism (Duncan et al., 2007; Strahan et al., 2007; Nielsen et al., 2017), updated here to include reactions for sulfur species. The GMI mechanism in GEOSCCM has been extensively evaluated for its stratospheric ozone-related photochemistry and transport in various model intercomparisons, including Stratosphere-troposphere Processes and their Role in Climate (SPARC) Chemistry Climate Model Validation (CCMVal), CCMVal-2, and the CCMI (SPARC-CCMVal, 2010; Eyring et al., 2010, 2013; Morgenstern et al., 2017). Aerosol species are simulated by the Goddard Chemistry, Aerosol, Radiation, and Transport, second generation (GOCART-2G), module (Collow et al. 2024), which includes a sectional approach for dust (five bins), sea salt (five bins), and nitrate (three bins), and a bulk approach for sulfate (dimethyl sulfide, $SO_2$, methanesulfonic acid, and $SO_4^{2-}$) aerosol and carbonaceous species (hydrophobic and hydrophilic modes of "white" and "brown" organics and black carbon).

For the GEOSCCM simulations performed with the GOCART-2G module we use the nominal GOCART-2G sulfate mechanism, updated here to use the online hydroxyl (OH) radical, nitrate ($NO_3$) radical, and hydrogen peroxide ($H_2O_2$) from the GMI mechanism instead of climatological fields provided from offline files (Collow et al., 2024). While not a full coupling to the GMI sulfur cycle it nevertheless allows the GOCART-2G sulfate mechanism to have the impact of the Hunga water vapor perturbation on the oxidants. A second "instance" of the GOCART-2G sulfate mechanism is run that is specifically for the volcanic $SO_2$ and resultant sulfate from the Hunga eruption. This allows us to track the eruptive volcanic aerosol separately from the nominal sulfate instance that sees mainly tropospheric sources. We assign this volcanic instance optical properties consistent with SAGE retrievals of the sulfate aerosol properties, using an effective radius of 0.4 microns. We find that 750 Tg of $H_2O$ is needed in the initial injection to provide a residual ~150 Tg of water in the stratosphere after a week. All other injection parameters follow the protocol. The model spinup was performed by "replaying" to the MERRA-2 meteorology (Gelaro et al. 2017), and is used throughout the **Exp2a** results. A MERRA-2 2012-2021 climatology of SST and sea ice fractions are used based on Reynolds et al. (2002).

## 4.6 GEOS/CARMA

A second configuration of the GEOSCCM, coupled to the sectional aerosol microphysics package CARMA, also simulated the eruption (GEOS/CARMA). This configuration is the same as above except for the aerosol package and its coupling to the GMI chemistry mechanism. For this version of GEOSCCM, we use the configuration of CARMA described in Case et al. (2023).

This configuration uses 24 size bins, spread logarithmically in volume between 0.25nm and
6.7μm in radius and simulates the nucleation, condensational growth, evaporation, coagulation,
and settling of sulfate aerosols in these simulations following the mechanism of English et al.
(2013). For these simulations, CARMA is fully coupled to the GMI sulfur cycle by the
production (i.e., oxidation of $SO_2$, evaporation of sulfate aerosols) and loss (i.e., nucleation and
condensation of sulfate aerosols) of sulfuric acid ($H_2SO_4$) vapor. Optical properties for the
CARMA aerosols are calculated based on the interactively calculated aerosol size distribution.
The same injection parameters for GEOSCCM described above are used by this configuration.
This model configuration contributed to **Exp2a** and "replayed" to MERRA-2 meteorology as
above.
**4.7 GSFC2D**
The NASA/Goddard Space Flight Center two-dimensional (2D) chemistry-climate model
(GSFC2D) has a domain extending from the surface to ∼92 km (0.002 hPa). The model has 76
levels, with 1 km vertical resolution from the surface to the lower mesosphere (60 km) and 2 km
resolution above (60-92 km). The horizontal resolution is 4° latitude, and the model uses a 2D
(latitude-altitude) finite volume dynamical core (Lin & Rood, 1996) for advective transport. The
model has detailed stratospheric chemistry and reduced tropospheric chemistry, with a diurnal
cycle computed for all constituents each day (Fleming et al., 2024). The model uses prescribed
zonal mean surface temperature as a function of latitude and season based on a multi-year
average of MERRA-2 data (Gelaro et al., 2017).  Zonal mean latent heating, tropospheric water
vapor, and cloud radiative properties as a function of latitude, altitude, and season are also
prescribed (Fleming et al., 2020).
For the free-running simulations, the model planetary wave parameterization (Bacmeister
et al., 1995; Fleming et al., 2024) uses lower boundary conditions (750 hPa, ∼2 km) of
geopotential height amplitude and phase for zonal wave numbers 1–4. These are derived as a
function of latitude and season using: 1) a 30-year average (1991–2020) of MERRA-2 data for
the standard yearly-repeating climatological-dynamics simulations ("Clim-NoQBO"); and 2)
individual years of MERRA-2 data (1980-2020) randomly rearranged in time to generate
interannual variations in stratospheric dynamics ("ensemble1", "ensemble2",…"ensemble10").
For the inter-annually varying dynamics simulations, the model includes an internally generated
QBO (Fleming et al., 2024).
For experiments that include the Hunga volcanic aerosols, the simulations go through the
end of 2023, using prescribed aerosol properties for 2022-2023 from both the GloSSAC data set
and derived from the OMPS-LP data (Taha et al., 2021, 2022). For experiments that include the
Hunga $H_2O$ injection, Aura/MLS observations are used to derive a daily zonal mean Hunga
water vapor anomaly in latitude-altitude, which is added to the baseline $H_2O$ (no volcano)
through the end of February 2022. This combined water vapor field is then fully model computed
starting 1 March 2022 through the end of 2031.
For **Exp2b**, the model zonal mean temperature and transport fields are computed from
the MERRA-2 reanalysis data. These are input into the model and used as prescribed fields (no
nudging is done).
**4.8 IFS-COMPO**
The Copernicus Atmosphere Monitoring Service (CAMS) provides daily global analysis
and 5-day forecasts of atmospheric composition (aerosols, trace gases, and GHGs) (Peuch et al.

2022). CAMS is coordinated by the European Centre for Medium Range Weather Forecasts (ECMWF) and uses, for its global component, the Integrated Forecasting System (IFS), with extensions to represent aerosols, trace, and GHGs, being called "IFS-COMPO" (also previously known as "C-IFS", Flemming et al. 2015). IFS-COMPO is composed of IFS(AER) for aerosols, as described in Remy et al. (2022) while the atmospheric chemistry is based on the chemistry module as described in Williams et al. (2022) for the troposphere (IFS-CB05) and Huijnen et al. (2016) for the stratosphere (IFS-CBA). The stratospheric chemistry module of IFS-COMPO is derived from the Belgian Assimilation System for Chemical ObErvations (BASCOE, Errera et al 2019). IFS-COMPO stratospheric chemistry is used since the operational implementation of cycle 48R1 on June 27, 2023 (Eskes et al., 2024).

The aerosol component of IFS-COMPO is a bulk aerosol scheme for all species except sea salt aerosol and desert dust, for which a sectional approach is preferred, with three bins for each of these two species. Since the implementation of operational cycle 48R1 in June 2023, the prognostic species are sea salt, desert dust, organic matter (OM), black carbon (BC), sulfate, nitrate, ammonium, and secondary organic aerosols (SOA).

For **Exp2a**, cycle 49R1 IFS-COMPO has been used, which will become operational for CAMS production in November 2024, at a resolution of TL511 (~40 km grid cell) over 137 model levels from surface to 0.01 hPa. Cycle 49R1 IFS-COMPO integrates a number of updates of tropospheric and stratospheric aerosols and chemistry. The most relevant aspect for this work concerns the representation of stratospheric aerosols, which has been revisited with the implementation of a coupling to the stratospheric chemistry through a simplified stratospheric sulfur cycle including nucleation/condensation and evaporation processes, as shown in **Figure 4**. Direct injection of water vapor into the stratosphere is expected to enhance the nucleation and condensation of sulfate through the reaction with $SO_3$ and production of gas-phase $H_2SO_4$.

The volcanic injection takes place between 3 and 6 UTC on January 15, 2022, with a uniform vertical distribution between 25 and 30 km of altitude, over a rectangular region of 400 km (latitude) x 200 km (longitude) centered on the coordinates of the Hunga volcano. The injected quantities are 0.5 Tg $SO_2$ and 190 Tg $H_2O$.

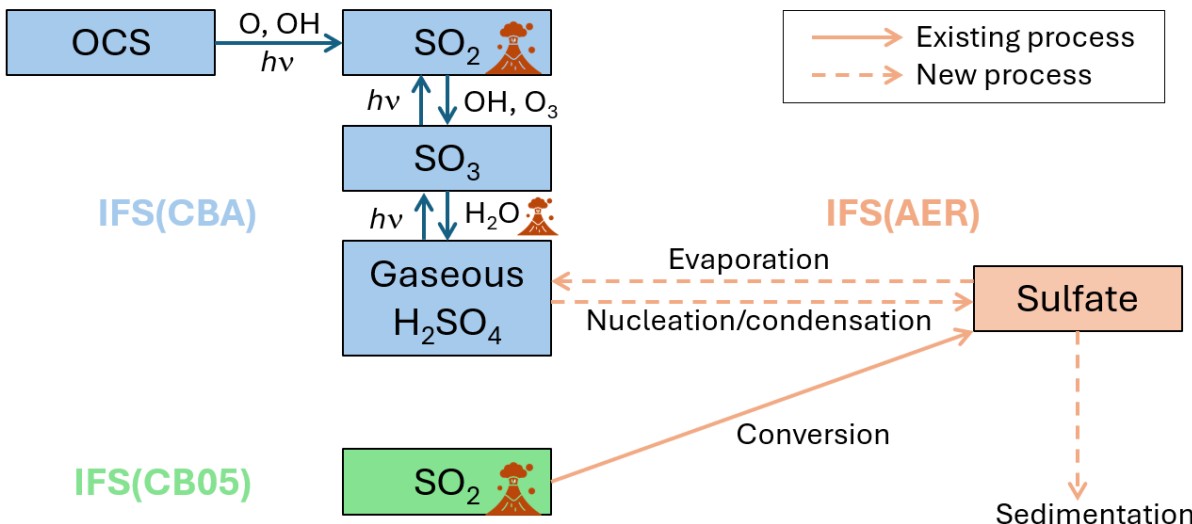

**Figure 4.** Architecture of the stratospheric extension of IFS(AER) and its coupling with IFS(CBA) and IFS(CB05), with existing and new processes implemented in cycle 49R1 of IFS-COMPO. *hν*

represents photolysis and the volcano symbols represent direct injections by volcanic eruptions.
Sedimentation is indicated as a new process because it has been revisited.
### 4.9 LMDZ6.2-LR-STRATAER and LMDZ6.2-LR-STRATAER-REPROBUS

The Institut Pierre-Simon Laplace Climate Modelling Centre (IPSL CMC, see
https://cmc.ipsl.fr) has set up a new version of its climate model in the runup of CMIP6. Further
description of the IPSL-CM6A-LR climate model can be found in Boucher et al. (2020) and in
Lurton et al. (2020). New development of the model is now ongoing to prepare the IPSLCM7
version.
The IPSLCM7 climate model is using the general circulation model named LMDZ for
*Laboratoire de Météorologie Dynamique-Zoom* (Hourdin et al., 2006). The LMDZ version used
for this study is based on a regular horizontal grid with 144 points regularly spaced in longitude
and 142 in latitude, corresponding to a resolution of $2.5° \times 1.3°$. The model has 79 vertical layers
and extends up to 80 km, which makes it a "high-top" model. The model shows a self-generated
quasi-biennial oscillation (QBO) whose period has been tuned to the observed one for the
present-day climate (Boucher et al., 2020).
The aerosol is interactively simulated in the STRATAER module using a sectional
scheme with 36 size bins. STRATAER is an improved version of the Sectional Stratospheric
Sulfate Aerosol (S3A) module (Kleinschmitt et al., 2017). It now takes into account the
photolytic conversion of $H_2SO_4$ into $SO_2$ in the upper stratosphere (Mills et al., 2005). The size-
dependent composition of $H_2SO_4/H_2O$ aerosols is now computed iteratively to ensure that the
surface tension, density, and composition are consistent in the calculation of the Kelvin effect.
The surface tension, density, $H_2SO_4$ vapor pressure, and nucleation rates are calculated based on
Vehkamäki et al. (2002). The version of the LMDZ6.2-LR-STRATAER atmospheric model used
in the HTHH Impact project accounts for the stratospheric $H_2O$ source from methane oxidation.
The chemistry is simulated using the REPROBUS (*REactive Processes Ruling the Ozone*
*BUdget in the Stratosphere*) chemistry module that includes 55 chemical species and a
comprehensive description of the stratospheric chemistry (Marchand et al., 2012, Lefèvre et al.,
1994, Lefèvre et al., 1998).
For **Exp2a**, the $H_2O$ and $SO_2$ is injected at 27.5 km altitude using a Gaussian distribution
and standard deviation of 2.5 km. The injection latitude ranges from 22°S to 14°S, and longitude
ranges from 182°E to 186°E. The injections of $H_2O$ and $SO_2$ are 150 Tg and 0.5 Tg, respectively.
The SSTs are taken from the IPSL climate coupled simulation run under the CMIP6 Tier 1
SSP2-4.5 scenario (Neil et al., 2016).
### 4.10 MIROC-CHASER

The Model for Interdisciplinary Research On Climate - CHemical Atmospheric general
circulation model for Study of atmospheric Environment and Radiative forcing (MIROC-
CHASER) version 6 (Sekiya et al. 2016) is a chemistry climate model, with a top at around 0.004
hPa. The present version of MIROC-CHASER is built on MIROC6 (Tatebe et al. 2019) and has a
spectral horizontal resolution of T85 (1.4° latitude × 1.4° longitude). The model has 81 vertical
levels, with a vertical resolution 0.7 km in the lower stratosphere, ~1.2 km in the upper stratosphere,
and ~3 km in the lower mesosphere. In the free-running simulations, the model generates
QBO internally. The ensemble members have different initial conditions (January 1, 2022), which
are generated using slightly different nudging relaxation time during the spin-up. The aerosols are

interactively simulated using a three-mode modal aerosol module (Seikiya et al. 2016). The chemistry uses comprehensive troposphere-stratosphere chemistry (Watanabe et al. 2011). The volcanic emission from continuously degassing volcanoes uses the emission inventory of Fioletov et al. (2022). For the explosive volcanic eruptions during the spin-up time, explosive volcanic emissions follow Carn (2022).

For **Exp1** fixed SST simulations, the model uses the observed SST from 10-year climatological mean from 2012 to 2021 using the montly-1deg CMIP6 AMIP SST (Gates et al., 1999).

For **Exp2a**, the atmospheric temperature and winds are nudged to MERRA-2 reanalysis with a 12-hour relaxation using 3-hour meteorology. The observed SST uses the NOAA 1/4° Daily Optimum Interpolation Sea Surface Temperature (OISST) from 2022 to 2023 (Huang et al. 2020).

The initial volcanic injection altitude and area are not tuned but follow the experimental protocol. For **Exp1** and **Exp2a**, the $H_2O$ and $SO_2$ are injected at 25 to 30 km altitude. The injection latitude ranges from 22°S to 14°S, and longitude ranges from 182°E to 186°E. The initial injection of $H_2O$ is 186 Tg, with ~150 Tg left after the first week following the eruption. The large initial $H_2O$ injection is necessary to keep 150 Tg in the stratosphere as requested by the experimental protocol, because a large amount of ice clouds generates and falls to the troposphere soon after the eruption.

## 4.11 UKESM1.1

The United Kingdom Earth System Model (UKESM, Sellar et al., 2019, 2020) is the successor to the HadGEM2-ES model (Collins et al., 2011), jointly developed by the UK Met Office and the Natural Environment Research Council (NERC) to deliver simulations to the Coupled Model Intercomparison Project Phase 6 (CMIP6; Eyring et al., 2016). For HTHH-MOC, we run the updated UKESM1.1 system (Mulcahy et al., 2023) which consists of the physical climate model HadGEM3-GC3.1 (Kuhlbrodt et al., 2018; Williams et al., 2018), and has improved tropospheric aerosol processes and aerosol radiative forcings (Mulcahy et al., 2018; 2020). The GC3.1 system comprises the GA7.1 global atmosphere model configuration (Walters et al., 2019), which uses the ENDGAME dynamics system (Wood et al., 2014), at a resolution of 1.875° longitude by 1.25° latitude with 85 levels extending to 85 km. Specifically the simulations apply the UKESM1.1-AMIP academic community release job (at v12.1 of the Unified Model), as supported by the UK National Centre for Atmospheric Science.

The interactive atmospheric chemistry module UKCA (UK Chemistry and Aerosols) has a number of chemistry configurations; with UKESM1.0 for CMIP6 applying the combined stratosphere and troposphere chemistry (CheST) option (Archibald et al., 2020), essentially a combination of the stratosphere chemistry (Morgenstern et al., 2009) and tropospheric chemistry (O'Connor et al., 2014) UKCA schemes. The UKCA aerosol scheme is the GLOMAP-mode aerosol microphysics module (Mann et al., 2010; 2012; Bellouin et al., 2013), with UKESM1.0 including the initial set of adaptations to GLOMAP for simulating stratospheric aerosol (Dhomse et al., 2014). For all UKESM1.0 integrations for CMIP6, the system was applied with evaporation of sulphate aerosol de-activated, stratospheric aerosol properties enacted from the CMIP6 prescribed zonal mean data set (Luo, 2017), but for the integrations here we have applied the system for interactive aerosol across the troposphere and stratosphere, enacting a Hunga emission of volcanic $SO_2$ following the 0.5Tg@25-30km Tonga-MIP protocols (see **Table 6**).

For the improved UKESM1.1 version applied here, the other most relevant development, compared to UKESM1.0 used for CMIP6, is the interactive atmospheric chemistry module UKCA

(UK Chemistry and Aerosols) has the updates to heterogeneous chemistry added by Dennison et
al. (2019), to represent more realistically reactions occurring on the surfaces of polar stratospheric
clouds and sulfate aerosol, with modified uptake coefficients of the five existing reactions and the
addition of a further eight reactions involving bromine species.    For these simulations, we have
added to UKESM for the first time the equilibrium liquid PSC scheme of Carslaw et al. (1995), an
interim implementation here coupling the 5 existing heterogeneous reactions chlorine activation
then occurring on both solid and now also liquid ternary-aerosol PSCs.
For **Exp2**, UKESM1.1 is run in specified dynamics configuration (Telford et al., 2008,
2009), the atmospheric temperature and winds nudged to ERA5 every 6 hours, the Newton
relaxation applied for levels 12 to 80 of 85 (between 1 km and 60 km)  Sea-surface temperatures
and sea-ice are prescribed from the Reynolds v2.1 datasets, both during the 2017 to 2022 spin-up
period, and the 2-year experiment 2 period to December 2023.   Monthly varying anthropogenic
atmospheric chemistry and aerosol emissions were set following the CMIP6 SSP2-4.5 datasets.

**4.12 WACCM6/MAM4**
The Whole Atmosphere Community Climate Model version 6 (WACCM6; Gettelman et
al. 2019) is the high-top version of the atmospheric component of the Community Earth System
Model, version 2 (CESM2), with a top at around 140 km. WACCM6 has a horizontal resolution
of 0.9° latitude × 1.25° longitude, utilizing the finite volume dynamical core (Lin & Rood,
1996). The model has 70 vertical levels, with a vertical resolution ~1 km in the lower
stratosphere, ~1.75 km in the upper stratosphere, and ~3.5 km in the upper mesosphere and lower
thermosphere (Garcia et al., 2017). In the free-running simulations, the model generates QBO
internally (Mills et al., 2017; Gettelman et al. 2019). The ensemble members differ in the last
date of nudging (from January 27 to February 5, 2022). The aerosol is interactively simulated
using a four-mode modal aerosol module (MAM4; Liu et al., 2012, 2016; Mills et al., 2016), in
which we used the Vehkamäki nucleation scheme (Vehkamäki et al., 2002). The chemistry uses
comprehensive troposphere-stratosphere-mesosphere-lower-thermosphere (TSMLT) chemistry
(Gettelman et al. 2019). The volcanic emissions from continuously degassing volcanoes use the
emission inventory of Andres and Kasgnoc (1998). For the explosive volcanic eruptions during
the spin-up time, explosive volcanic emissions follow Mills et al. (2016) and Neely III and
Schmidt (2016) with updates until 2022.
For **Exp1_CoupledOcean** simulations, the ocean and sea-ice are initialized on January
3, 2022 with output from a standalone ocean model forced by atmospheric state fields and fluxes
from the Japanese 55-year Reanalysis (Tsujino et al., 2018). To accurately simulate the early
plume structure and evolution, the winds and temperatures in WACCM are nudged toward the
Analysis for Research and Applications, MERRA-2 meteorological data (Gelaro et al., 2017)
throughout January 2022. After February 1, 2022, the model is free-running to capture fully-
coupled variability. For the fixed SST simulation, the model uses the 10-year climatology SST
from 2012 to 2021. The SST data is OISSTv2, which is a NOAA High-resolution (0.25x0.25)
Blended Analysis of Daily SST and Ice (Banzon et al., 2022).

For **Exp2**, the atmospheric temperature and winds are nudged to MERRA-2 reanalysis with a 12-hour relaxation using 3-hour meteorology (Davis et al., 2022). The observed SST uses 10-year climatological mean from 2012 to 2021.

The initial volcanic injection altitude and area are the same as described for section 4.1 CAM5/CARMA.

### 4.13 WACCM6/CARMA

WACCM6/CARMA only performed **Exp2** and used a configuration similar to WACCM6/MAM4 with the same horizontal and vertical resolution, SSTs, and meteorological nudging. Differences compared to WACCM6/MAM4 are the chemistry and aerosol configuration used. WACCM6/CARMA used the middle atmosphere chemistry with limited chemistry in the troposphere and comprehensive chemistry in the stratosphere, mesosphere and lower thermosphere (Davis et al., 2022). Furthermore, we use the Community Aerosol and Radiation Model for Atmospheres (CARMA, Tilmes et al. 2023, based on Yu et al., 2015 with some updates) as the aerosol module, in which we used the Vehkamäki nucleation scheme (Vehkamäki et al., 2002). CARMA defines 20 mass bins and tracks the dry mass of the particles and assumes particle water is in equilibrium with the environmental water vapor. The approximate radius ranges from 0.2 nm to 1.3 μm in radius for the pure sulfate group that sulfate homogeneous nucleation occurs in, and ranges from 0.05 to 8.7 μm in the mixed group that tracks all major tropospheric aerosol types (i.e. black carbon, organic carbon, sea salt, dust, sulfate).

The initial volcanic injection altitude and area are determined by validating the water and aerosol transportation in the first six months against MLS and OMPS observations. In these simulations, the $H_2O$ is injected to 25 to 35 km altitude following Zhu et al. (2022), while the $SO_2$ is injected 82% of the total mass to 26.5-28 km and 18% to 28-36 km altitude. The injection latitude ranges from 22°S to 6°S, and longitude ranges from 182.5°E to 202.5°E.

## 5. Preliminary results

The models' performances will be evaluated focusing on the following aspects: the stratospheric aerosol optical depth will be compared with GloSSAC and other satellite instruments individually such as OMPS-LP, SAGEIII-ISS, and OSIRIS; the aerosol effective radius will be compared with balloon observations (Asher et al., 2024), SAGEIII-ISS retrieved size distribution and AeroNet retrieved particle radius; the water vapor lifetime, ozone and its related chemicals (such as HCl, HNO3, CLO) will be compared with MLS observations; the temperature anomaly will be compared with MLS detrended temperature field (Randel et al., 2024). All the evaluations

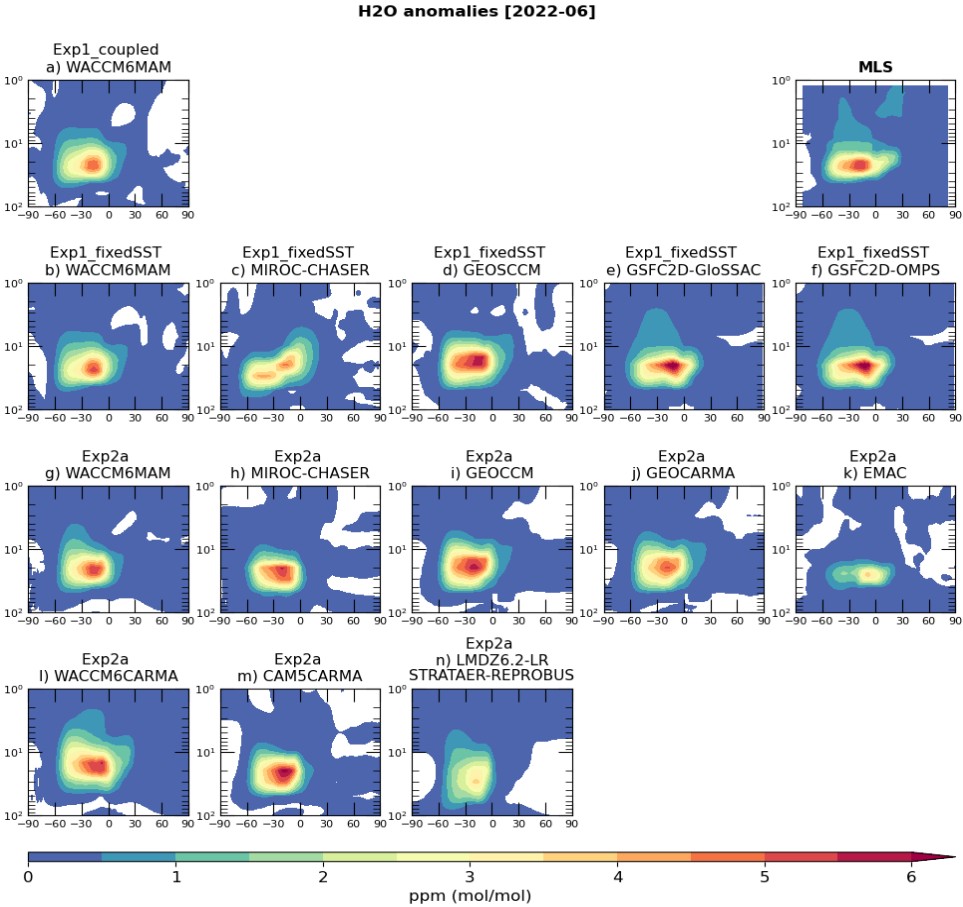

**Figure 5.** the zonal average $H_2O$ anomaly in June 2022 from MLS, Exp1_fixedSST, Exp1_CoupledOcean and Exp2a. The simulated anomaly is using $H_2O+SO_2$ run minus the control run. And the MLS uses the 2022 data minus the climatology.

will be conducted before looking into the climate impact of this eruption, such as radiative impact and tropospheric responses. This work will be described in a follow up manuscript.

As this manuscript is written, we are still completing the model output inspection and validation phase. So, we can only provide preliminary results from some models. **Figure 5** shows the preliminary results from Exp1 and Exp2 in June 2022 compared with the MLS v5 water vapor anomaly. The model results shown here generally agree with MLS anomaly regarding the vertical (10-50 hPa) and horizontal distribution (60˚S to 20˚N), and the anomaly peaking at ~ 6 ppmv for most of the models. This consistency of water vapor anomaly six months after the eruption helps

us have confidence in these models on the analysis of climate and chemistry impacts, and will be
evaluated in detail in the follow up studies.
**6. Summary**
A multi-model observation comparison project is designed to evaluate the impact of the
2022 Hunga eruption. Two experiments are designed to cover various research interests for this
eruption, including sulfate and water plume dispersion and transport, dynamical and chemical
responses in the stratosphere, and climate impact. The project will not only benefit the Hunga
Impact assessment, but also benchmark the model performance on simulating stratospheric
explosive volcanic eruption events and stratospheric water vapor injections. These events have a
potentially large impact on the Earth system, especially on the stratospheric ozone layer and
radiative balance.
**Code/Data availability**
The data used to produce the results used in this paper is archived on Zenodo: Wang, X. (2025).
MLS H2O anomaly 2022 [Data set]. Zenodo. https://doi.org/10.5281/zenodo.14962954; Quaglia,
I., Aquila, V., & Zhuo, Z. (2025). HTHHMOC zonal average H2O anomaly in June 2022 from
multi-model data [Data set]. Zenodo. https://doi.org/10.5281/zenodo.14963276; Andrin, J.
(2025). REMAP-GloSSAC-2023 [Data set]. Zenodo. https://doi.org/10.5281/zenodo.14961868;
Brühl, C. (2025). SO2 emission for EMAC MPIC model [Data set]. Zenodo.
https://doi.org/10.5281/zenodo.14962925.
**Author Contributions:**
Y.Z. Concept design, Project Administration, Experiment design, data archive, WACCM models
setup;
E.A. provides NOAA balloon aerosol and water vapor observations for experiments
E.B. and S.T. and J.Z. Experiment design, conducts experiments using WACCM6MAM;
A.B. Experiment design, Data archive;
A.J. Experiment 2b prescribed fields preparation;
M.K. provides GloSSAC data for Exp 2b;
Takashi S. and S.W.: S.W. conducted all MIROC-CHASER experiments, data post-processing,
data archive under supervision of Takashi S., who developed the aerosol microphysics scheme of
the model.
X.W. and W.Y. Conducts experiment using WACCM6MAM;
Z.Z. Conducts experiment using WACCM6MAM, WACCM6MAM data post-processing, data
archive;
N.L. and S.B.: Conducts experiment using IPSL7-STRATAER, data post-processing and archive
M.M. and L.F.: Conducts experiment using IPSL7-STRATAER-REPROBUS,  data post-
processing and archive
S.R. and S.C. Conducts experiments using IFS-COMPO
M.C. Experiment design, Tonga-MIP lead;
F.F.Ø., G.K., O.M. contributed to experiment design
C.B. Conducts experiment using EMAC
I.Q., V.A., R.U. and A.K. Model output inspection and evaluation
E.F. Conducts experiments using GSFC2D, data post-processing, and data archive.
D.P. Contributed to experiment design and conducted experiments using CMAM and data post-
processing
P.R.C., L.D.O., Q.L., M.M., and S.S. Contributed to experiment design and conducted
experiments with the NASA GEOS CCM
P.C. and P.R.C. Contributed to experiment design and conducted experiments with the NASA
GEOS CARMA model
H.A. and Y.Y. Conducts experiment using CCSRNIES-MIROC3.2, data post-processing and
archive
D.V. contributed to experiment design and assisted E.B. with variables request
W.R. and P.N. concept design
G.M. concept design and in charge of JASMIN data archiving
P.Y. and Y.P. conduct experiments using CAM5CARMA and data post-processing
S.T. and C.-C. L. conduct experiments using WACCM6CARMA and data post-processing
**Competing interests**
We declare at least one of the co-authors is on the editorial board of GMD.

**Acknowledgement:**
We acknowledge Michelle Santee, Martyn Chipperfield, Allegra Legrande, Thomas Peter,
Myriam Khodri for their valuable input for this project.
We acknowledge the APARC for their funding and other support on this activity.
This research has been supported by the National Oceanic and Atmospheric Administration
(grant nos. 03- 01-07-001, NA17OAR4320101, and NA22OAR4320151). NCAR's Community
Earth System Model project is supported by the National Center for Atmospheric Research,
which is a major facility sponsored by the NSF under Cooperative Agreement No. 1852977.
W.Y.'s work was performed under the auspices of the U.S. Department of Energy by Lawrence
Livermore National Laboratory under Contract DE-AC52-07NA27344. TS and SW were
supported by MEXT-Program for the advanced studies of climate change projection (SENTAN)
Grant Number JPMXD0722681344 and their MIROC-CHASER and MIROC-ES2H simulations
were conducted using the Earth Simulator at JAMSTEC. IFS-Compo is supported by the
Copernicus Atmosphere Monitoring Service (CAMS), which is one of six services that form
Copernicus, the European Union's Earth observation programme.
The IPSLCM7 model experiments were performed using the high-performance computing
(HPC) resources of TGCC (Très Grand Centre de Calcul) under allocations 2024-A0170102201
(project gen2201) provided by GENCI (Grand Équipement National de Calcul Intensif). This
study benefited from the ESPRI (Ensemble de Services Pour la Recherche l'IPSL) computing
and data centre (https://mesocentre.ipsl.fr) which is supported by CNRS, Sorbonne Université,
École Polytechnique and CNES.
V.A. is supported by the NASA NNH22ZDA001N-ACMAP and NNH19ZDA001N-IDS
programs.
F.F.Ø. acknowledge support from the European Union's Horizon 2020 research and innovation
programme under the Marie Skłodowska-Curie grant 891186.
R.U. is supported by NASA Upper Atmospheric Composition Observations and Aura Science
Team programs as well as through the NASA Internal Scientist Funding Model.
P.R.C., L.D.O., Q.L, S.S., M.M., and P.C. are supported by the NASA Modeling Analysis and
Prediction program (program manager: David Considine, NASA HQ) through the NASA

Internal Scientist Funding Model. P.C. is additionally supported by the NASA Postdoctoral
Program. GEOS CCM and GEOS CARMA simulations were performed at the NASA Center for
Climate Simulation.

H.A. and Y.Y were supported by KAKENHI (JP24K00700 and JP24H00751) of the Ministry of
Education, Culture, Sports, Science, and Technology, Japan, and NEC SX-AURORA
TSUBASA at NIES were used to perform CCSRNIES-MIROC3.2 simulations.

GC acknowledges funding from the European Commission via the ERC StG 101078127 and the
Spanish Ministry of Science and Innovation via the Ramon y Cajal grant no. RYC2021-033422-
I.

We acknowledge funding from the UK National Centre for Atmospheric Science (NCAS)
for Graham Mann and Sandip Dhomse via the NERC multi-centre Long-Term Science
programme on the North Atlantic climate system (ACSIS, NERC grant NE/N018001/1.

A.J. acknowledges support from the Swiss National Science Foundation (SNSF) project AEON
(grant no. 200020E_219166).

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
