# Peer review of "Hunga Tonga-Hunga Ha'apai Volcano Impact Model 1"

_EGUsphere, 2024_

## Author Response (AR1)

Dear authors,

Unfortunately, after checking your manuscript, it has come to our attention that it does not comply with our "Code and Data Policy".

https://www.geoscientific-model-development.net/policies/code_and_data_policy.html

First, the site that you link to provide access to the GloSSAC data (the Atmospheric Science Data Center from NASA) is not a suitable repository for scientific publication, as it does comply with the minimum requirements for long-term archival and data preservation. Therefore, you must provide a link and permanent identifier (e.g. DOI) to one of the acceptable repositories that we list in our policy, and that contain the GloSSAC data you have used for your work.

Second, for the figures 1 and 3 in your manuscript you use MLS and OSIRIS data; however, in your Data Availability statement you do not provide information about the repository where it is possible to find the data for such figures. Please, address it and add the required information (link to the repository and permanent identifier) for the repositories containing the MLS and OSIRIS data necessary to replicate the figures.

Juan A. Añel

Geosci. Model Dev. Executive Editor

Thank you for your valuable comments. We modify the data availability:

GloSSAC: NASA/LARC/SD/ASDC. (n.d.). Global Space-based Stratospheric Aerosol Climatology Version 2.22 [Data set]. NASA Langley Atmospheric Science Data Center DAAC. Retrieved from https://doi.org/10.5067/GLOSSAC-L3-V2.22.

Processed GloSSAC for model use: Jörimann, A., 2024. REMAP-GloSSAC-2023 [dataset]. ETH Zürich, url: http://hdl.handle.net/20.500.11850/713396. doi: 10.3929/ethz-b-000713396.

MLS data: Lambert, A., Read, W. and Livesey, N. (2020), MLS/Aura Level 2 Water Vapor (H2O) Mixing Ratio V005, Greenbelt, MD, USA, Goddard Earth Sciences Data and Information Services Center (GES DISC), Accessed: 01/2024–04/2024, 10.5067/Aura/MLS/DATA2508.

The complete OSIRIS data set can be downloaded from https://arg.usask.ca/docs/osiris_v7/. Rieger, L. A., Zawada, D. J., Bourassa, A. E., and Degenstein, D. A.: A Multiwavelength Retrieval Approach for Improved OSIRIS Aerosol Extinction Retrievals, J. Geophys. Res.-Atmos., 124, 2018JD029897, https://doi.org/10.1029/2018JD029897, 2019.

We sincerely appreciate your time and effort to give us valuable suggestions for this paper. We prepared the answers below.

This paper focuses on the description of simulations pertaining to the understanding of the Tonga-Hunga eruption impacts on stratospheric conditions, radiative forcing and climate impacts. The paper is currently quite confusing and as such does not provide a clear understanding on how the results can be used for the stated goals. I have the following comments that the authors should address before submitting a revised version of the paper

1) As far as I can tell, there is not really an experiment #4. It is just only for models that have a representation of the mesosphere. Unless the authors can better justify the reason to have a specifically defined experiment, it should be removed

The experiment#4 is combined in the description of Exp1 now and in all the tables.

2) In table 5, what is exp 1/4? This is never discussed in the paper

Since Exp4 is not there, now exp1/4 is exp1.

3) Lines 146-148: the paper describes a set of experiments designed to isolate the specific roles of SO2 and H2O. How do those map with the numbering of the experiments? This is not clear at all.

Each experiment (Exp1, Exp2) needs to conduct a set of runs including: a) control. b) H2O&SO2; c) H2O; d) SO2 (optional).

We reword line 167-169. Hopefully it is clearer now. "There are two experiments (**Exp1** and **Exp2** detailed below) designed to fulfill the scientific goals. Each experiment includes a set of simulations with different volcanic injections (i.e. with and without water and/or SO2 injections),"

4) It is confusing if exp 3 is the same as Tonga-MIP or not. If it is the same, then why is there an exp 3? If it is not the same, then all details should be provided. By the way, the reference to Margot Clyne's thesis is not a valid way to allow readers to access this information. Also, it seems that there is a lot of freedom in the way modeling groups will perform those simulation (fixed SSTs or not, free running or nudged). How will it be possible to intercompare when so many things are different? Is there the assumption that those won't matter? Why not following exp 1 design?

We now list TongaMIP as a parallel MIP activity for the assessment to avoid confusion. We add explanations to explain why TongaMIP is a necessary independent from Exp1 and 2.

Line 129: "Two multi-model evaluation projects are designed to facilitate the success of this activity: Tonga Model Intercomparison Project (Tonga-MIP) (Clyne et al. 2024) and the Hunga Tonga-Hunga Ha'apai Volcano Impact Model Observation Comparison (HTHH-MOC) Project (this paper)."

Line 319: "A parallel model intercomparison project Tonga-MIP (Clyne et al., 2024) will also be part of the 2025 Hunga assessment, which is designed to explore the plume evolution between 1 day and up to 1 or 2 months after the eruption. Tonga-MIP was initiated before the APARC Hunga Activity started. It will be described in a separate paper, but we list it in this paper to document the comprehensiveness of the modeling effort for the Hunga assessment. Two purposes of Tonga-MIP cannot be achieved by Exp1 and 2: 1. The nudged experiment of Tonga-MIP aims to intercompare the microphysics processes (i.e., cloud and aerosol physics and sulfur chemistry) between different models. Therefore, all models are requested to inject 150 Tg of water, but the retaining of the water varies between models, differing from Exp1 and 2, which ask to retain ~150 Tg of water in the stratosphere. SO2 injection is 0.5 Tg, the same as experiments in HTHHMOC. The injections are required to be injected between 25-30 km, within the latitude and longitude box of 22-14°S and 182-186°E, at a constant vertical volume mixing ratio for 6 hours starting at 4 UTC on January 15th. 2. The free-run experiment of Tonga-MIP aims to study the radiative effect of water and $SO_2$ on the Hunga plume descending and ascending during the first month after the eruption since the Hunga water and aerosol plumes were observed to descend several kilometers during the first monthly after the eruption (Sellito et al., 2022; Randel et al., 2024). Therefore, Tonga-MIP designed to nudge the atmosphere up until several different dates and explore the plume descending patterns with free-run atmosphere after these dates. The dates are Jan 21, Jan 26 and Jan 31. Most of the models that participate in Tonga-MIP also participate in the HTHH-MOC."

5) If I follow the timeline, results have been available for a few months already by the time this paper was submitted.  It would be useful to have some basic results that would show the usefulness of these experiments

We added a preliminary result to section 5. However, as of this manuscript is written, we are still undergrounding the model output inspection and validation phase. So, we can only provide preliminary results from a few models.

Line 850 and Figure 5: "As this manuscript is written, we are still completing the model output inspection and validation phase. So, we can only provide preliminary results from some models. **Figure 5** shows the preliminary results from Exp1 and Exp2 in June 2022 compared with the MLS v5 water vapor anomaly. The model results shown here generally agree with MLS anomaly regarding the vertical (10-50 hPa) and horizontal distribution (60˚S to 20˚N), and the anomaly peaking at ~ 6 ppmv for most of the models. This consistency of water vapor anomaly six months after the eruption helps us have confidence in these models on the analysis of climate and chemistry impacts, and will be evaluated in detail in the follow up studies."

[Figure]

**Figure 5.** the zonal average $H_2O$ anomaly in June 2022 from MLS, Exp1_fixedSST, Exp1_CoupledOcean and Exp2a. The simulated anomaly is using $H_2O+SO_2$ run minus the control run. And the MLS uses the 2022 data minus the climatology.

6) The authors should clarify what they mean by radiative forcings. Some of the model descriptions include the mention that these are performed as double calls to the radiation. Is that the protocol followed by everyone?

The model produces the instantaneous radiative effect (with double call) and adjusted radiative effect. We change the radiative forcing to radiative effect in the paper.

7) In my opinion, it is very different to run a model with fixed SSTs or coupled ocean/sea-ice. Therefore those should not be lumped into the same experiment 1 naming. That will make the analysis very confusing

We renamed the Exp1 to Exp1_FixedSST and Exp1_CoupledOcean. And it will be a separate folder and this name is specified in time series of the file name. The modification of the paper is in Table 2 and Table 5.

Minor comments

1) the references line 806-812 are not lined up properly

Reference lining fixed.

2) Line 292: those models should be included here as well

Since Tonga-MIP now is listed as a separate MIP activity, we delete this part to avoid confusion.

3) Line 208: is it adopted or adapted?

Exp3 is deleted.

4) The goal of nudging (lines 182-184) is not so much to reduce interannual variability as to ensure that the meteorology will be as close as possible to the one observed.

Fixed.

5) Line 191: typo sS

Fixed.

6) Why is Fig. 3 shown? If it is something relevant to the general discussion, it should be moved to the main section, not the model description

Figure 3 is to show the specific $SO_2$ retrieval that EMAC MPIC used in their model set up. We add in the caption: "The $SO_2$ injection used in EMAC MPIC model is based on the Hunga $SO_2$ perturbation derived from extinction observed by OSIRIS".

We sincerely appreciate your time and effort to give us valuable suggestions for this paper. We prepared the answers below.

The manuscript presents the overview of the multi-model intercomparison project to investigate the impacts of the HTHH volcanic eruption on atmospheric dynamics, chemistry, and climate with different experiment designs. I think the research goals and questions expected to be answered in this paper are scientifically significant. However, several areas need to be clarified or justified. Description of the analyzation and evaluation of model outputs could be included.

Major comments:

1. The scope and relevance of Exp4 to the overall project goals are relatively unclear compared with other experiments. It feels more like a sub-Exp affiliated to Exp1 according to Table 2 & 5.

   The experiment#4 is combined in the description of Exp1 now and in all the tables.

2. The manuscript lacks a more detailed description of how models' performance will be evaluated and compared with each other.

   Line 777, we add: "The models' performances will be evaluated focusing on the following aspects: the stratospheric aerosol optical depth will be compared with GloSSAC and other satellite instruments individually such as OMPS-LP, SAGEIII-ISS, and OSIRIS; the aerosol effective radius will be compared with balloon observations (Asher et al., 2024), SAGEIII-ISS retrieved size distribution and AeroNet retrieved particle radius; the water vapor lifetime, ozone and its related chemicals (such as HCl, HNO3, CLO) will be compared with MLS observations; the temperature anomaly will be compared with MLS detrended temperature field (Randel et al., 2024). All the evaluations will be conducted before looking into the climate impact of this eruption, such as radiative impact and tropospheric responses. This work will be described in a follow up manuscript."

3. The data sources of SST are not clear for different experiments and models. Some model descriptions contain SST data source, but some don't. For example, Lines 594-595 only state that MIROC-CHASER for Exp1 will use 10-year mean of observed SST but didn't mention which dataset is used; In terms of Exp2, MIROC-CHASER uses NOAA OISST (Line: 598) data but WACCM6 uses so-called 10-year climatological SST mean (Line: 634). I think authors should provide more complete information about data sources and initial/boundary conditions for all models and experiments.

   We added the ocean data source for the models that needs an ocean state.

   WACCM uses the same ocean data, line 802: "The SST data is OISSTv2, which is a NOAA High-resolution (0.25x0.25) Blended Analysis of Daily SST and Ice."

4. Meanwhile, how to compare simulated results if they use different data inputs and initial/boundary conditions? Partitioning sources of uncertainty might be needed.

The reviewer is correct in that modelling studies of climate change projections often partition the sources of uncertainty into the three main categories: internal variability, model uncertainty, and scenario uncertainty (Hawkins and Sutton, 2009; Lehner et al. 2020). However, we do not believe such an approach is explicitly required in this case. First, the role of internal variability is accounted for by using a combination of nudged and free-running simulations, and the use of multiple ensemble members (minimum 10) in case of the latter. Regarding model uncertainty, while we didn't ask for the model to use the same source of ocean state in the fixed SST runs, an observed ocean state should be used which should also be the same for the control and perturbed simulations; this means the ocean state of each model should be fairly close and hence could effectively be incorporated into the model uncertainty. Finally, only one scenario uncertainty is assessed in this intercomparison, i.e. the scenario with the Hunga eruption compared to the control simulation without it. We note that in the projection of stratospheric volcanic forcing, we only considered the Hunga eruption since 2022, and no future explosive eruptions are included (for example, the 2024 Mt. Ruang eruption contributed to elevated stratospheric aerosol optical depth, but it is not included).

Line 207 add "Note that in the projection of stratospheric volcanic forcing, we only considered the Hunga eruption since 2022, and no future explosive eruptions are included. For example, the 2024 Mt. Ruang eruption contributed to elevated stratospheric aerosol optical depth, but it is not included."

Minor comments:

1. Line 104: the 'aerosol optical depth' is a singular noun. It should not be followed by 'are'.

Corrected.

2. A figure plotting some preliminary results might be helpful.

We added a figure 5 of current model that had done the data validation.

Line 790: "As this manuscript is written, we are still completing the model output inspection and validation phase. So, we can only provide preliminary results from some models. **Figure 5** shows the preliminary results from Exp1 and Exp2 in June 2022 compared with the MLS v5 water vapor anomaly. The model results shown here generally agree with MLS anomaly regarding the vertical (10-50 hPa) and horizontal distribution (60°S to 20°N), and the anomaly peaking at ~ 6 ppmv for most of the models. This consistency of water vapor anomaly six months after the eruption helps us have

confidence in these models on the analysis of climate and chemistry impact, and will be
evaluated in detail in the follow up studies."

[Figure]

**Figure 5.** the zonal average $H_2O$ anomaly in June 2022 from
MLS, Exp1_fixedSST, Exp1_CoupledOcean and Exp2a. The
simulated anomaly is using $H_2O+SO_2$ run minus the control
run. And the MLS uses the 2022 data minus the climatology.

3. I am not sure if Table 4 is important enough to take a whole page in the main content. I
would rather to see a table listing output variables.

A table of output variables are listed in Supplementary Excel Table. But we forgot to
upload it during the first submission. It will be included in the second revision.

4. I assume 'Exp1/4 (coupled ocean)' in Table 5 (Line: 302) refers to 'Exp1 and Exp4 with
the coupled ocean simulation (Line: 624)' but it should be identified earlier to avoid
confusion.

Now Exp4 is combined with Exp1. So we modified this part according too.

---

## Author Response (AR2)

We corrected all the technique questions raise below.

I think the revised manuscript has demonstrate its significance and provided detailed description of the experiment design. I noticed some textual elements (words and/or numbers) are partially covered by Figure5, which need to be corrected.

The authors have adequately addressed comments raised in the previous review steps. I recommend the manuscript to be published in GMD after considering the following technical corrections:
Table 2: Please try to avoid the unhyphenated breaks of words in the table. (And why is the number of ensemble members indicated as "-" for Exp2, instead of "1"?)
l.282: "HTHHMOC-Exp1and4" may be outdated as there is no Exp4 anymore in the revised version, as I understand? Anyway, I find the detailed description of the file naming convention including several examples unnecessary for the paper body text.
Table 4: "clyne et al." -> "Clyne et al."
l.410: What does "between boundary layer and 100 hPa" mean? Above the boundary layer? Or all levels between surface and 100 hPa?
l.414 is there a reference, url or doi that can be given?